# Missed Opportunities for Vaccination and Associated Factors among Children Attending Primary Health Care Facilities in Cape Town, South Africa: A Pre-Intervention Multilevel Analysis

**DOI:** 10.3390/vaccines10050785

**Published:** 2022-05-16

**Authors:** Chukwudi A. Nnaji, Charles S. Wiysonge, Abdu A. Adamu, Maia Lesosky, Hassan Mahomed, Duduzile Ndwandwe

**Affiliations:** 1Division of Epidemiology and Biostatistics, School of Public Health and Family Medicine, University of Cape Town, Cape Town 7925, South Africa; charles.wiysonge@mrc.ac.za (C.S.W.); maia.lesosky@uct.ac.za (M.L.); 2Cochrane South Africa, South African Medical Research Council, Cape Town 7501, South Africa; abdu.adamu@gmail.com (A.A.A.); duduzile.ndwandwe@mrc.ac.za (D.N.); 3Division of Epidemiology and Biostatistics, Department of Global Health, Stellenbosch University, Cape Town 7505, South Africa; 4HIV and Other Infectious Diseases Research Unit, South African Medical Research Council, Cape Town 7501, South Africa; 5Western Cape Provincial Department of Health, Cape Town 8000, South Africa; hassan.mahomed@westerncape.gov.za; 6Division of Health Systems and Public Health, Department of Global Health, Stellenbosch University, Cape Town 7505, South Africa

**Keywords:** missed opportunities for vaccination, vaccination, immunisation, children, primary health care, quality improvement, South Africa

## Abstract

Despite the substantial efforts at ensuring universal access to routine immunisation services among children in South Africa, major gaps in immunisation coverage remain. This study assessed the magnitude of missed opportunities for vaccination (MOV) and associated factors among children aged 0–23 months attending primary health care (PHC) facilities in Cape Town. We used multilevel binomial logistic regression models to explore individual and contextual factors associated with MOV, with children aged 0–23 months at Level 1, nested within PHC facilities (Level 2). A total of 674 children and their caregivers were enrolled. MOV prevalence was 14.1%, ranging from 9.1% to 18.9% across sub-districts. Dose-specific MOV prevalence was highest for the second dose of measles vaccine (9.5%) and lowest for the first dose of rotavirus vaccine (0.6%). The likelihood of a child experiencing MOV was significantly associated with caregivers’ low level of education (Odds ratio (OR) = 3.53, 95% credible interval (CrI): 1.13–11.03), recent receipt of immunisation messages (OR = 0.46, 95%CrI: 0.25–0.87), shared immunisation decision making by both parents (OR = 0.21, 95%CrI: 0.07–0.62) and health facility staff number (OR = 0.18, 95%CrI: 0.06–0.61). The burden of MOV among children in Cape Town is influenced by individual and contextual factors, which provide important opportunities for quality improvement and broader strategies to improve routine immunisation service delivery.

## 1. Introduction

Immunisation has been recognised as one of the greatest advances in public health and a highly cost-effective means of disease prevention [1,2,3]. Through the Expanded Programme on Immunisation (EPI), more than 100 million infants are immunised every year, helping to avert millions of child deaths due to vaccine-preventable diseases annually [4,5,6]. Immunisation also has enormous societal and economic benefits through the aversion of catastrophic healthcare costs and the lost productivity due to ill health [7,8]. Despite the substantial progress made in expanding immunisation coverage globally since the launch of the EPI in 1974, coverage has stalled in recent years and remains suboptimal in many countries [9]. South Africa in particular is struggling to increase its national immunisation coverage. This is despite the country’s significant efforts towards ensuring universal access to routine immunisation services [10,11]. According to recent World Health Organization (WHO) and United Nations Children’s Fund (UNICEF) estimates, immunisation coverage remains sub-optimal (i.e., below 90%) across all routine vaccine doses in South Africa [10]. These gaps are likely to have been exacerbated by the COVID-19 pandemic’s disruption of essential health services, including routine immunisation [12,13,14,15] Sub-optimal immunisation coverage among children in South Africa has been attributed to individual-level factors, such as low awareness of the importance of immunisation and low socioeconomic status, and health system factors, such as human resource shortage, health workers’ immunisation training gaps and vaccine stock-outs, among others [6,7,8,9,10,11,12,13,14,15,16,17,18,19].

The WHO has recognised missed opportunities for vaccination (MOV) as a major contributor to sub-optimal immunisation coverage globally [20,21]. To accelerate and sustain immunisation coverage progress, the WHO recommends the provision of immunisation services at every contact with the health system [20]. A missed opportunity for vaccination refers to any contact with health services by an individual who is eligible for vaccination (unvaccinated or not up-to-date, and free of contraindications to vaccination), which does not result in the individual receiving all the vaccine doses for which s/he is eligible [22]. The prevalence of MOV varies widely across countries, with an average of 32.2% in low- and middle-income countries (LMICs) [23]. In sub-Saharan Africa, the prevalence ranges from as high as 89% in Gabon to 16% in Swaziland [24,25]. We hypothesise that the burden is of substantial magnitude at national and sub-national levels in South Africa.

Common enabling factors of MOV have included the failure or inability of health providers to screen patients for eligibility, perceived contraindications to vaccination on the part of providers and parents, vaccine shortages and rigid clinic schedules that separate immunisation from other health services [22]. Individual-level factors found to be associated with MOV include child’s age and birth order, maternal age, level of education, media access and attendance of antenatal, and household income [24,26,27,28,29]. Contextual and health system factors, such as area of residence, health facility proximity, number of available health workers/vaccinators and unavailability of vaccines, have also been found to be associated with MOV [25,30,31].

To better understand the burden of MOV and its structural and contextual determinants, there have been increasing MOV assessment efforts globally [32,33,34]. Research efforts in this area have particularly grown over the last decade, including in African and other LMIC contexts [24,25,30,32,33,35]. However, not much is known about the MOV burden and determinants in the South African context [16,36]. Findings from our recent analysis of secondary data from the 2016 South African Demographic and Health Survey data suggest a significant MOV burden, with a national prevalence of 42% and Western Cape provincial prevalence of 36% among children aged 12–23 months [37] The study identified individual-level factors, such maternal attendance of antenatal care during pregnancy, and geographical factors, such as province of residence, as significant determinants of MOV.

The current study used a globally standardised tool to assess the prevalence and factors associated with MOV among children aged 0–23 months attending PHC facilities in Cape Town. This study represents a baseline assessment, the findings of which will help inform the design and implementation of a facility-level quality improvement programme to address identified MOV. The findings will help identify actionable and sustainable quality improvement strategies and interventions that can be scaled up across districts, provinces and nationally to strengthen immunisation service delivery and close the current coverage gaps. Further details of the study rationale and an overview of its methodological approach have been published elsewhere [38].

## 2. Materials and Methods

### 2.1. Study Design

A cross-sectional, multi-level modelling approach was used to assess the burden of MOV and factors associated with it across individual (child and caregiver) and contextual (sub-district and facility) levels. The multi-level analytical design helped account for the hierarchical nature of the data and potential clustering of individual-level characteristics within health facilities.

### 2.2. Study Setting

This study was conducted in PHC facilities across five health sub-districts within the Cape Town Metropole. The Cape Town Metropolitan Health District has eight legislated sub-districts serving a population of 4.1 million persons [39]. Like elsewhere in South Africa, routine immunisation services in Cape Town are funded through the Expanded Programme on Immunisation of South Africa (EPI-SA) and provided free of charge primarily through the PHC facilities [40]. The current routine immunisation schedule for children in South Africa is outlined in Table 1 below.

### 2.3. Study Population

Caregivers of children aged 0–23 months attending PHC facilities in the Cape Town Metro for any type of health service on the day of assessment were included in this study. To be eligible for inclusion, the child must have been brought to the PHC facility by an adult caregiver (aged ≥ 18 years). If a caregiver visited the hospital with two or more children, the youngest child’s data was primarily collected to avoid overrepresentation of caregivers with multiple children.

### 2.4. Sample Size Estimation

The WHO MOV assessment methodology recommends a minimum sample size of 600 child–caregiver pairs [22,32]. For this study, a minimum size of 630 was estimated based on a MOV prevalence of 32.2% (based on findings of a previous systematic review assessing the burden of MOV in low-and-middle income countries) [23,41], an alpha level of 0.05 (95% confidence interval), a margin of error of 5%, non-response rate of 20% and a design effect of 1.5 [23,42,43]. A design effect adjustment was made to account for the clustering of respondents within specific health facilities and the stratified nature of the sampling.

### 2.5. Sampling

Using a modified one-stage cluster sampling technique, five health sub-districts within the Cape Metro Health District were randomly selected. From these sub-districts, 11 PHC facilities were selected using convenience sampling based on institutional approval. Each selected PHC facility served as a cluster, from which all eligible and consenting caregivers with a child aged 0–23 months were consecutively recruited until study sample size was reached.

### 2.6. Data Collection

Study data were collected and managed using Research Electronic Data Capture (REDCap) electronic data capture tool hosted at the University of Cape Town [44]. Data were collected through exit interviews using an electronic structured and interviewer-administered questionnaire (see Appendix A). The questionnaire was adapted from the WHO tool for assessing MOV in health care settings [45] and pre-tested. The questionnaire has 6 sections: Appendix A relates to data on the child; Appendix A on parent/caregiver; Appendix A on the use of the Road to Health Booklet (RtHB) for checking vaccination status and information on vaccine administered; Appendix A on current visit; Appendix A on quality of the vaccination service; and Appendix A on reasons for vaccination. Exit interviews were conducted by trained data collection research assistants who were fluent in English, Afrikaans and IsiXhosa. Data collectors positioned themselves at the main exits of each health facility, approached caregivers accompanied by children as they were exiting the facility and asked if they were willing to participate in the study.

### 2.7. Variables

**Outcome variable**: MOV was defined based on the World Health Organization (WHO) definition as a binary variable of whether or not a 0–23-month-old child had any contact with a PHC facility but remained unvaccinated with any vaccine doses for which the child was eligible [22]. This definition was operationalised by using the child’s date of birth and date at which the last vaccine doses were administered to determine if a child was up to date with immunisation based on the South Africa national childhood immunisation schedule and, if not, which vaccine doses had been missed. Children who were fully immunised for age and free of contraindications at the time of interview were categorised as not having MOV, while others who had missed at least one dose for which they were eligible were deemed to have MOVs.

**Explanatory variables**: These were stratified along across two levels:**Level 1 (individual-level factors).** These included child-related factors (such as age of child, sex of child, birth order and birth weight) and caregiver-related factors (such as caregiver relationship with child, marital status, mothers’ attendance of antenatal care, level of education, employment status, mode of transport to health facility, duration of transport to health facility and recent exposure to immunisation messages);**Level 2 (health-facility-level factors):** These included facility-related factors, such as facility size (clinic vs. community health centre (CHC)), location (sub-district), number of staff, patient load, vaccine availability or stock-outs and immunisation scheduling.

### 2.8. Data Analysis

Continuous variables, such as age, were expressed as mean and standard deviations, while categorical variables were expressed as frequencies and percentages. Pearson’s Chi-square statistic was used to explore the distribution of MOV across categorical individual- and facility-level variables. Due to the hierarchical nature of the data collected, a multilevel analytical approach to the multivariable modelling was used [46]. As an extension of generalised linear models, multilevel models help address the clustering of data by generating cluster-specific random models [47]. In this study, a two-level binomial logistic regression modelling approach was adopted, given the binary nature of the outcome variable (MOV), with individual (child and parent/caregiver-related factors) at level 1, all nested within PHC facilities (at level 2). Four models were fitted. Model 1 was an empty (null) model with no explanatory variable; Model 2 contained only individual-level (child and caregiver) factors; Model 3 contained only facility-level factors; and Model 4 (full model) controlled for individual- and facility-level factors.

The models were fitted using the Bayesian Markov Chain Monte Carlo (MCMC) method [48,49]. In this method, a Markov chain generates successive samples of parameters from their posterior distributions [48,49]. In this study, the MCMC model estimation settings were specified to achieve a burn-in period of 10,000 iterations followed by a monitoring period of 5000 iterations. From the posterior estimates, the results of fixed effects (measures of association) were reported as odds ratios (ORs) with their 95% credible intervals (CrIs). For the random effects, variance, intraclass correlation coefficients (ICC) and median odds ratios (MOR) were measured to quantify the attributable influence of health facility factors. The ICC represents the percentage of the total variance in the odds of MOV attributable to the context (health facility), serving as a measure of clustering of the odds of MOV in the same health facility. Relatedly, MOR represented the total variance in MOV probability that was attributed to health facilities on the odds ratio scale. The deviance information criterion (DIC) was used to assess model goodness of fit, with lower DIC indicating a better fit [48]. Multi-collinearity was assessed among explanatory variables using the variance inflation factor (VIF). Multi-level analysis was performed using MLwiN version 3.01 and Stata 14.2, using the *runmlwin* command [50,51].

## 3. Results

### 3.1. Participants’ Characteristics

A total of 674 children aged 0–23 months and their caregivers nested within 11 PHC facilities across five health sub-districts in the Cape Town metropolis were enrolled. The mean age (±SD) of the children was 7.5 (±1.2) months. Most (92.5%) of the caregivers were mothers, with mean age of 29.3 (±6.9) years. Immunisation visits accounted for a majority (79.3%) of the children’s health facility visits. Nearly all (99.6%) parents/caregivers brought their children’s Road to Health Booklets (RtHB) to the clinic on the survey day. The distribution of respondents across socio-demographic and contextual variables is shown in Table 1.

### 3.2. Facility Characteristics

All of the PHC facilities provide immunisation services either every day or most days. Health workers routinely asked to see children’s RtHB (on 98.2% of the encounters). The majority (62.8%) of the health facilities have fewer than 50 health workers. Immunisation waiting time was 30 min or longer in 70.0% of the facilities. About a fifth (22.4%) of the facilities experienced vaccine stock-outs, while a minority (5.7%) experienced a vaccine cold-chain disruption in the past three months. Of the caregivers whose children were vaccinated on survey day, less than half (49.2%) were informed of the vaccines being administered during immunisation encounters, while only 16.8% received information on the possible side effects of vaccines administered to their children.

### 3.3. Prevalence of MOV

After excluding children who were up to date at the time of exit interview (110) and those with incomplete records (4), the remaining 561 children were included as denominator for the MOV analysis. Of these, 79 (14.1%) experienced MOV in at least one eligible vaccine dose. Figure 1 below illustrates the estimation of MOV prevalence. Variations in the prevalence of MOV were observed across a number of individual- and facility-level variables.

As shown in Table 2, MOV prevalence ranged from 9.1% in the Western health sub-district to 18.9% in the Khayelitsha sub-district. The prevalence of MOV was higher among children whose mothers did not attend antenatal care during pregnancy (50.0%), whose caregivers had low (primary) level of education (26.9%), who were in the health facilities for non-vaccination services (20.4%), whose primary caregivers did not receive any immunisation message in the last three months (19.6%) and those whose caregivers walked to the health facility (i.e., did not have own transport) (17.1%).

### 3.4. Dose-Specific MOV Prevalence

As shown in Table 3, the prevalence of MOV was highest for the second dose of the measles vaccine (9.5%), fourth dose of the hexavalent vaccine containing diphtheria, tetanus, pertussis, inactivated polio, *Haemophilus influenzae* type b and hepatitis B antigens (DTaP-IPV-Hib-HepB) (8.5%) and second dose of the oral polio vaccine (OPV) (6.5%). It was lowest for the first dose of the pneumococcal conjugate vaccine (PCV) (1.0%), first dose of the rotavirus vaccine (RV) (0.6%) and first dose of the DtaP-IPV-Hib-HepB vaccine (0.6%).

### 3.5. Measures of Association (Fixed Effects)

Table 4 shows the results of different models. In the fully adjusted model (Model 4), after controlling for the effects of individual- (child and caregiver) and contextual (facility)-level factors, caregiver level of education, receipt of immunisation messages, shared immunisation decision making by both parents and facility staff number were significantly associated with missed opportunities for vaccination. Children whose caregivers had only a primary level of education were more likely to experience MOV than those whose caregivers had a higher (post-primary) level of education (OR = 3.53, 95%CrI: 1.13–11.03). Conversely, the odds of MOV were lower among children whose caregivers received immunisation messages in the past three months (OR = 0.46, 95%CrI: 0.25–0.87) than those whose caregivers did not receive immunisation messages. Similarly, children whose parents were both involved in shared decision making concerning their immunisation were less likely to have MOV than those whose immunisation decision making was not a shared responsibility between both parents (OR = 0.21, 95%CrI: 0.07–0.62). Another notable finding was that children visiting facilities with high staff numbers (50 or more personnel) experienced lower odds of MOV relative to those visiting facilities that were less human resourced (OR = 0.18, 95%CrI: 0.06–0.61).

### 3.6. Measures of Variations (Random Effects)

As shown in Table 4, in Model 1 (unconditional model), there was a substantial variation in the odds of MOV across facilities (variance = 0.49, 95%CrI: 0.20–1.15). According to the intra-facility correlation coefficient, 12.85% of the variance in odds of MOV could be attributed to facility-level factors. Results from the median odds ratio (MOR) are also in keeping with facility-level factors influencing children’s odds of experiencing MOV. From the full model (Model 4), it was estimated that if a child moved to another facility with a higher probability of MOV, their odds of MOV would increase by 2.03-fold. The DIC for Model 4 was 366.17.

## 4. Discussion

### 4.1. Main Findings

The prevalence of missed opportunities for vaccination among children attending primary health care facilities in Cape Town was found to be 14.1%, ranging from 9.1% to 18.9% across sub-districts. Dose-specific prevalence of MOV was highest for the second dose of measles vaccine (9.5%) and lowest for the first dose of rotavirus vaccine (0.6%) and first dose of DTaP-IPV-Hib-HepB vaccine (0.6%). After controlling for the effects of individual- and contextual-level variables, factors such as caregiver level of education, receipt of immunisation messages, shared immunisation decision making by both parents and facility staff number were found to be independently associated with the odds of MOV among children.

### 4.2. Implications for Immunisation Practice and Quality Improvement

Although the burden of MOV prevalence found in this study is relatively lower than the prevalence reported in previous studies conducted in other LMIC and African settings [23,32,35], it is of substantial magnitude, which may undermine current efforts to improve immunisation coverage among children in Cape Town and South Africa in general. It is worthy of note that the prevalence observed in the current study is lower than the Western Cape province-wide prevalence of 36% found in our previous study using a South African national dataset [37], while it is higher than the 5% reported by another study in Cape Town [16]. It could be that these disparities are due to the fact that these previous studies were based on MOV assessment methodologies, evaluation settings and time frames different from those of the current study. Overall, the observed prevalence represents a substantial gap that warrants attention and efforts to address it and improve routine immunisation coverage among children in Cape Town and South Africa.

Although many caregivers reported that health workers had asked to see their children’s Road to Health Booklets (RtHB), we found that health workers do not always screen the booklets for children’s immunisation status at every visit. Interventions to foster routine immunisation status checks at all visits can quickly identify children with missed vaccine doses. It is thus important that health facilities and staff are equipped with the appropriate training, tools, standard operating procedures and resources for routine screening of children’s immunisation eligibility and the provision of catch-up vaccination to eligible children. This could involve strategies such as placing reminder tags on children’s RtHBs at the point of registration, placing posters/charts with MOV information prominently in consulting rooms and routine supervision to improve RtHB screening compliance. Our finding that dose-specific prevalence of MOV was higher for the second dose of the measles and the fourth dose of the DTaP-IPV-Hib-HepB vaccines implies that immunisation status screening may be particularly important for vaccines given much later in childhood, which, according to the current and previous studies, tend to experience higher odds of MOV [32]. To ensure synergy between vaccination and other services, all health workers, including immunisation and non-immunisation personnel, must be able to correctly screen children’s immunisation status and eligibility at all service delivery points. This is especially crucial for mitigating the rate of MOV in children accessing non-immunisation services, given the evidence that the likelihood of MOV may be higher in such settings.

Our study suggests that human resource constraints may be a determining factor of MOV, with MOV being more prevalent in facilities with fewer staff. This is consistent with evidence from previous studies showing that shortage of healthcare staff at the facility level is a major contributor to MOV [52,53]. This may lead to long immunisation waiting times [53,54]. In settings of staff shortages, health facilities may consider leveraging non-clinical staff (such as administrative and security personnel) to assist with non-clinical aspects of vaccination services, such as instituting a ‘triage’ system that identifies children who are eligible for vaccination through RtHB screening at the point of entry or waiting rooms. Alternatively, an exit-screening strategy may be instituted to identify children with incomplete vaccination status as they are exiting the health facility and have them referred immediately to receive outstanding doses. These can help alleviate some of the pressure on the clinical staff.

The finding that MOV was associated with caregivers’ access to immunisation messages was consistent with reports of previous studies [55,56]. There is substantial evidence that immunisation-themed health education interventions targeted at parents can increase not only parents’ knowledge and intention to vaccinate but also children’s vaccination uptake and immunisation coverage [55,56]. Such education interventions, when tailored to the target audience’s needs, such as in local languages, may be particularly effective and useful for addressing enablers of MOV, such as parental or caregiver low level of education, as found in this study. Immunisation education interventions can be implemented at both the facility level, through health talks in waiting rooms, antenatal and postnatal clinic sessions, and at the community level, through regular campaigns, particularly in communities with low childhood vaccination rates.

Another notable finding of the current study was the role of shared parental immunisation decision making as a determinant of MOV. Similar to previous findings, our study points to the importance of parental involvement in shared decisions about children’s health [57,58]. Interventions aimed at promoting joint parental decision making regarding children’s immunisation have been described. One such intervention was aimed at fostering the involvement of fathers as co-decision makers with mothers. It improved immunisation rates and the percentage of parents intending to vaccinate their infants [58]. Moreover, it is important to address existing parental or caregiver information gaps, such as those identified in our study, to effectively share parental immunisation decision making for reducing MOV. Parents require access to good-quality immunisation information (such as information on the vaccines being administered, available options, health benefits and risks of possible side effects) to make informed decisions on behalf of their child.

In line with the WHO MOV assessment strategy, which seeks to answer three important questions: ‘what is the burden of MOV, what are the reasons for the missed opportunities, and what can be done to address these?’, this study constitutes an initial step in the broader MOV strategy [45]. It offers important baseline findings and pre-intervention lessons for informing evidence-based and locally appropriate interventions to reduce the identified burden of MOV. The findings underscore the need to implement practical and context-specific interventions to reduce MOV burden, with the ultimate aim of contributing to improved immunisation coverage by leveraging existing routine immunisation delivery systems [45,59]. Specifically, we identified significant opportunities to improve immunisation service delivery through quality improvement strategies, standardising catch-up policies and vaccination checks, addressing health worker constraints, improving knowledge surrounding vaccination schedules and contraindications, and ensuring that necessary vaccination supplies are available.

### 4.3. Implications for Broader Policy and Practice

While much of the burden of MOV will be amenable to facility-level quality improvement and other local interventions, broader immunisation programme and health system actors, such as policy makers, also have their important roles to play, particularly in addressing the structural factors underlying MOV. First, health and EPI policy makers need to consider policies that will enhance the efficient use of home-based records (HBRs), such as the RtHBs, for screening children’s immunisation status. This can be through policy frameworks for regular in-service training to strengthen health workers’ competences in home-based records screening, prompt administration of and catch-up immunisation where children with missing doses have been identified. Similarly, it is important to update health workers’ training on multi-dose vial policies to address common concerns of vaccine wastage if children are vaccinated outside of the dedicated facility immunisation days or sessions. Furthermore, efforts to harmonise RtHBs across provincial and sub-national health systems can help enhance the effectiveness of RtHB screening by health workers in settings such as Cape Town that receive a substantial number of migrants from other provinces.

The issue of human resource constraints, which was identified as a significant determining factor of MOV, also requires due attention. Health system and immunisation programme authorities at the national or sub-national level must make the necessary efforts for health facilities to be adequately staffed. This can be achieved through regular performance reviews to identify and address human resources gaps in a timely manner. In addition to increasing the number of staff where that is feasible, it is also important that existing health care staff are well knowledgeable and competent about immunisation and other primary care services. For instance, gaps in the knowledge of indications for and valid contraindications to immunisations, which have been reported as contributing factors to MOV, can be addressed through regular in-service training [32]. Where resource constraints make it unrealistic to increase staffing, efforts should be made to safeguard existing immunisation human resource capacity. Furthermore, immunisation service delivery could be re-aligned for a more efficient use of available staff, such as through task-shifting strategies, such as the training of non-clinical staff and lay health works to perform vaccination status checks.

In the present study, we found that low educational attainment was associated with higher odds of MOV. This is consistent with findings from previous studies. The available evidence has unequivocally shown children of parents or primary caregivers with low level of education were more likely to experience MOV [24,25,30]. Given that educational attainment is a correlate of socioeconomic status, this finding also suggests that, consistent with previous evidence in the literature, children from socioeconomically disadvantaged backgrounds may be more likely to experience MOV. These underscore the importance of system-wide efforts to address the underlying social determinants of health. In addition, specific measures, such as locally adapted health education and immunisation messaging, tailored to the needs of vulnerable populations, such as socioeconomic disadvantaged communities, may help overcome barriers posed by factors such as parental low level of education.

Notably, the emergence of the COVID-19 pandemic and its far-reaching disruption of health services have posed further constraints on immunisation service delivery globally, and South Africa is no exception [12]. While the government has recommended that immunisation services continue uninterrupted during the lockdown period, there are indications that the pandemic has had negative impacts on essential health services, including routine childhood immunisation [13,14,15]. Thus, assessing MOV to identify children who might have been missed during the pandemic and instituting remedial actions have become more imperative for mitigating the pandemic’s further disruption of immunisation services [21,60]. This can help ensure that every encounter a child has with the health system is an opportunity to identify children with missed vaccine doses and for immediate catch-up vaccination. Such efforts will also boost the implementation and mainstreaming of existing national primary health care and immunisation service improvement mandates [61,62].

Overall, health system and EPI programme managers and policy makers have a crucial role in ensuring the integration of MOV assessments into the health system as a routine process to monitor and track immunisation programme performance. Similarly, MOV remedial measures should be an integral part of the process, either through facility-level or broader system-wide QI initiatives. It is important that the plans are well incorporated into District Health Plans, EPI programmatic work plans and facility-level patient flow to ensure sustainability.

### 4.4. Limitations and Strengths

Our study and its methodological approach are not without limitations. First, given the cross-sectional nature of this assessment, it is difficult to infer any causal relationship between MOV and its associated factors. However, the relationships have been consistently demonstrated across a wide array of previous MOV assessment studies. Secondly, the cross-sectional design also makes the participants’ responses prone to recall and social desirability biases. However, these were minimised by using children’s RtHBs to validate immunisation-status-related responses. Thirdly, this study involved children aged 0–23 months and focused on primary care facilities in the public healthcare sector. These limit the generalisability of study findings and their implications for all children and all healthcare settings in the reference populations. Nonetheless, this age group restriction is consistent with the methodological approach recommended by the WHO for MOV assessment [22]. Moreover, the majority of the population in Cape Town is dependent on the public sector for immunisation services, and they are the most vulnerable to the impact of vaccine preventable diseases. While the study used a sample size consistent with that recommended by the WHO, the limited sample size also poses a constraint on the representativeness of the data. Another important limitation of this study is that some health workers might have been aware of the MOV assessment and might have modified their immunisation practices during the assessment period. As such, it is possible that our study underestimated the true magnitude of MOV. However, we ensured that that only district managers were made aware of the actual outcome being investigated.

### 4.5. Implications for Future Research

Considering the above-mentioned limitations, it is imperative that future MOV assessments take into account a number of methodological considerations. First, further research in this direction can assess MOV at a national level to give more nationally representative and generalisable findings. Given the focus of the current study on public primary health care facilities located within urbanised metropolitan sub-districts, there remain unanswered questions regarding the burden and factors associated with MOV in secondary and tertiary care settings, as well as in facilities in the private and rural sectors in South Africa. These need to be explored through wider assessments involving facilities across all care levels and both sectors, spanning both rural and urban settings. It would also be valuable for future studies to assess MOV in non-facility settings, such as mobile clinics and community health outreaches, where the burden of MOV may be more substantial. Lastly, the use of qualitative and quantitative methods in understanding the magnitude and determinants of MOV cannot be over-emphasised. Hence, mixed-methods MOV assessment designs are especially recommended. To enhance the uptake of assessment findings for improving routine immunisation and facility practice, such research efforts should be embedded as part of integrated MOV assessment and remedial implementation research initiatives.

## 5. Conclusions

Evidence suggests a substantial level of MOV among children in primary health care settings in Cape Town. Individual- and contextual-level factors associated with MOV reflect the multi-dimensional nature of its underlying factors, underscoring the need for remedial measures to not only target individuals but also take into account socioeconomic and structural factors that enable MOV. Findings and lessons learned from the current study are vital for informing locally responsive quality improvement interventions to reduce MOV among children in primary and broader healthcare settings, which may be a practical strategy for improving immunisation coverage at the population level.

## Figures and Tables

**Figure 1 vaccines-10-00785-f001:**
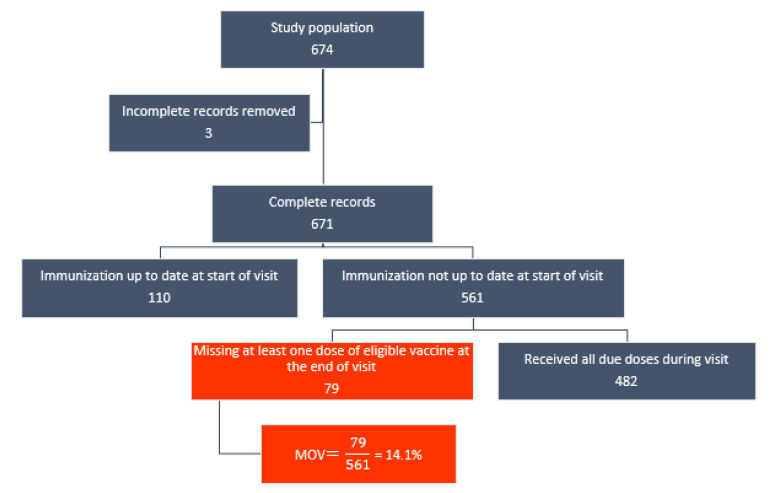
Flowchart of MOV prevalence estimation among children aged 0–23 months in Cape Town.

**Table 1 vaccines-10-00785-t001:** Current routine childhood immunisation schedule in South Africa.

Age	Vaccine Offered
Birth	BCG, OPV (0)
6 Weeks	OPV (1), RV (1), DTaP-IPV-Hib-HepB (1), PCV (1)
10 Weeks	DTaP-IPV-HIB-HepB (2)
14 Weeks	RV (2), DTaP-IPV-Hib-HepB (3), PCV (2)
6 Months	Measles (1)
9 Months	PCV (3)
12 Months	Measles (2)
18 Months	DTaP-IPV-Hib-HepB (4)
6 Years	Td (1)
9 Years	HPV (1), HPV (2) (2 doses, 6 months apart) *
12 Years	Td (2)

BCG = Bacille Calmette Guerin, DTaP-IPV-Hib-HepB = hexavalent vaccine (containing diphtheria, tetanus, pertussis, inactivated polio, *Haemophilus influenzae* type b and hepatitis B vaccines), HPV = human papillomavirus vaccine, OPV = oral polio vaccine, PCV = pneumococcal conjugate vaccine, RV = rotavirus vaccine, Td = tetanus and reduced dose diphtheria vaccine. * HPV vaccine is given as part of the school health programme rather than the EPI-SA.

**Table 2 vaccines-10-00785-t002:** Prevalence of missed opportunities for vaccination among children aged 0–23 months attending primary healthcare facilities in Cape Town.

Variables	Number of Children (%)	MOV Prevalence (%)	*p*-Value#
All children	561	79 (14.1)	
**Sub-district**			
Southern	99 (17.8)	9 (9.1)	
Western	38 (6.8)	5 (13.2)	
Tygerberg	240 (43.2)	32 (13.3)	
M/Plain	89 (16.0)	15 (16..9)	
Khayelitsha	90 (16.2)	17 (18.9)	0.346
**Facility type**			
CHC	269 (48.4)	36 (13.4)	
Clinic	287 (51.6)	42 (14.6)	0.671
**Facility ownership**			
Province	213 (38.2)	29 (13.6)	
City	345 (61.8)	49 (14.4)	0.902
**Vaccine stock-out in the past 3 months**			
Yes	125 (22.4)	17 (13.6)	
No	432 (77.6)	61 (14.1)	0.883
**Vaccine cold-chain challenges in the past months**			
Yes	32 (5.7)	2 (6.3)	
No	526 (94.3)	76 (14.5)	0.194
**Number of health workers**			
Fewer than 50	349 (62.8)	56 (16,1)	
50 or more	207 (37.2)	10 (10.63)	0.075
**Immunisation waiting time**			
Less than 30 min	147 (30.4)	14 (9.5)	
30 min or longer	336 (69.6)	51 (15.1)	0.070
**Child-level factors**			
**Age of child**	Mean (SD): 7.5 (1.2) months		
0–11 months	420 (75.3)	62 (14.8)	
12–23 months	138 (24.7)	16 (11.6)	0.148
**Sex of child**			
Female	280 (49.9)	33 (11.8)	
Male	281 (50.1)	45 (16.0)	0.148
**Birth order**			
1st–3rd order	501 (89.3)	67 (13.4)	
4th+ order	60 (10.7)	11 (18.3)	0.294
**Birth size**			
Large	141 (25.4)	23 (16.3)	
Average	351 (63.2)	47 (13.9)	
Small	63 (11.4)	7 (11.1)	0.556
**Reason for visit**			
Vaccination	445 (79.3)	55 (12.4)	
Non-vaccination	116 (20.7)	23 (19.8)	0.038
**Time of visit**			
Morning	342 (61.7)	48 (14.0)	
Afternoon	212 (38.3)	30 (14.2)	0.970
**Caregiver-related factors**			
**Caregiver age**	Mean (SD): 29.3 (6.9) years		
18–24 years	153 (27.4)	24 (15.7)	
25–34 years	281 (50.4)	38 (13.5)	
35+ years	124 (22.2)	16 (12.9)	0.764
**Level of education**			
Primary	26 (4.6)	7 (26.9)	
Post-primary	535 (95.4)	71 (13.3)	0.049
**Relationship to child**			
Mother	519 (92.5)	71 (13.7)	
Other relation	42 (7.3)	7 (16.7)	0.591
**Marital status**			
Not Married	369 (65.8)	56 (15.2)	
Married	192 (34.2)	22 (11.5)	0.227
**Maternal antenatal care**			
Attended	546	75 (13.7)	
Never attended	6	3 (50.0)	0.034
**Employment status**			
Employed	138 (30.0)	19 (13.8)	
Unemployed	423 (70.0)	59 (14.0)	0.958
**Means of transport**			
Own vehicle	61 (11.0)	3 (4.9)	
Public transport	178 (32.0)	21 (11.8)	
Walk	318 (57.0)	54 (17.0)	0.027
**Child immunisation message in the last 3 months**			
Yes	341 (61.3)	36 (10.6)	
No	215 (38.7)	42 (19.5)	0.003
**Child immunisation decision making**			
Both parents	117 (20.1)	10 (8.6)	
Not both parents	444 (79.1)	65 (15.3)	0.060

**Table 3 vaccines-10-00785-t003:** Antigen/dose-specific missed opportunities for vaccination among children aged 0–23 months in Cape Town.

**Age of Eligibility**	**Vaccine Antigen (Dose)**	**Number Eligible**	**MOV (%)**
	All doses	561	79 (14.1)
Birth	BCG	555	9 (1.6)
	OPV (0)	555	15 (2.7)
6 weeks	OPV (1)	520	34 (6.5)
	RV (1)	520	3 (0.6)
	DtaP-IPV-Hib-HepB (1)	520	3 (0.6)
	PCV (1)	520	5 (1.0)
10 weeks	DtaP-IPV-Hib-HepB (2)	439	6 (1.4)
14 weeks	RV (2)	381	14 (3.7)
	PCV (2)	381	12 (3.2)
	DtaP-IPV-Hib-HepB (3)	381	10 (2.6)
6 months	Measles (1)	287	12 (4.2)
9 months	PCV (3)	205	6 (2.9)
12 months	Measles (2)	137	13 (9.5)
18 months	DtaP-IPV-Hib-HepB (4)	59	5 (8.5)

**Table 4 vaccines-10-00785-t004:** Factors associated with missed opportunities for vaccination among children aged 0–23 months attending primary healthcare facilities in Cape Town.

	Model 1 ^a^ OR (95% CrI)	Model 2 ^b^ OR (95% CrI)	Model 3 ^c^ OR (95% CrI)	Model 4 ^d^ OR (95% CrI)
**INDIVIDUAL-LEVEL FACTORS**				
**Age of child**				
0–11 months		ref		ref
12–23 months		0.58 (0.27–1.25)		0.61 (0.28–1.35)
**Sex of child**				
Male		ref		ref
Female		0.63 (0.35–1.14)		0.70 (0.39–1.28)
**Birth order**				
1st–3rd order		ref		ref
4th+ order		1.83 (0.71–4.74)		2.05 (0.79–5.35)
**Birth size**				
Average		ref		ref
Non-average		0.89 (0.58–1.37)		0.74 (0.47–1.16)
**Reason for visit**				
Non-vaccination		ref		ref
Vaccination		0.55 (0.61–2.11)		0.69 (0.27–1.76)
**Time of visit**				
Morning		Ref		Ref
Afternoon		1.13 (0.61–2.11)		1.10 (0.58–2.07)
**CAREGIVER-LEVEL FACTORS**				
**Maternal age**				
18–24 years		ref		ref
25+ years		0.73 (0.45–1.16)		0.69 (0.43–1.11)
**Maternal education**				
Post-primary		ref		ref
Primary		3.03 (1.00–9.2)		3.53 (1.13–11.03)
**Marital status**				
Not Married		ref		ref
Married		0.92 (0.45–1.86)		1.03 (0.50–2.14)
**Maternal antenatal care**				
No				
Yes		0.51 (0.95–2.70)		0.57 (0.11–3.01)
**Maternal employment**				
Employed		ref		ref
Unemployed		1.29 (0.65–2.57)		1.33 (0.66–2.68)
**Means of transport to facility**				
Own vehicle		ref		ref
Public transport		1.49 (0.89–2.50)		1.51 (0.88–2.60)
**Child immunisation message in the last 3 months**				
No		ref		ref
Yes		0.45 (0.25–0.84)		0.46 (0.25–0.87)
**Child immunisation decision making**				
Other		ref		ref
Both parents		0.28 (0.10–0.80)		0.21 (0.07–0.62)
**FACILITY-LEVEL FACTORS**				
**Facility type**				
Clinic			ref	ref
CHC			0.71 (0.31–1.62)	0.72 (0.27–1.94)
**Facility ownership**				
Province			ref	ref
City			1.95 (1.18–3.23)	1.71 (0.90–3.24)
**Vaccine cold-chain disruption in the last 3 months**				
No			ref	ref
Yes			0.57 (0.12–2.72)	0.18 (0.02–1.71)
**Vaccine stock-out in the last 3 months**				
No			ref	ref
Yes			1.35 (0.42–4.30)	1.73 (0.42–7.16)
**Number of health workers**				
Fewer than 50			ref	ref
50 or more			0.33 (0.02–0.13)	0.18 (0.06–0.61)
**Random-effect estimates**				
Variance (95%CrI)	0.49 (0.20–1.15)	0.49 (0.18–1.34)	0.15 (0.01–11.42)	0.54 (0.01–4.27)
ICC (%)	12.85	12.87	4.46	14.27
MOR (%)	1.94	1.94	1.45	2.03
**Model fit statistics**				
DIC	445.87	365.26	447.38	366.17

^a^ Model 1—empty null model, without any explanatory variables (unconditional model). ^b^ Model 2—adjusted for only individual-level (child and caregiver) factors. ^c^ Model 3—adjusted for only contextual (facility)-level factors. ^d^ Model 4—full model, adjusted for individual- and facility-level factors. OR—odds ratio, CrI—credible interval, MOR—median odds ratio, DIC—Bayesian Deviance Information Criteria.

## Data Availability

The data presented in this study are available on request from the corresponding author.

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
