# Peer review of "Missed Opportunities for Vaccination and Associated Factors among Children Attending Primary Health Care Facilities in Cape Town, South Africa: A Pre-Intervention Multilevel Analysis"

_vaccines, 2022, doi:10.3390/vaccines10050785_

Round 1

Reviewer 1 Report

The paper is a very comprehensive study assessing missed opportunities for vaccination and associated factors among children attending primary health care facilities in Cape Town, South Africa. It was well conducted and the data obtained, obviously under difficult circumstances, was remarkable for what it revealed. The conclusions are appropriate, as are the stated shortcomings.

On line 306 there is an unnecessary question mark.

Author Response

Comment: The paper is a very comprehensive study assessing missed opportunities for vaccination and associated factors among children attending primary health care facilities in Cape Town, South Africa. It was well conducted and the data obtained, obviously under difficult circumstances, was remarkable for what it revealed. The conclusions are appropriate, as are the stated shortcomings.

Response: We are grateful for this acknowledgment of the strength and importance of our manuscript.

Comment: On line 306 there is an unnecessary question mark.

Response: Thank you for pointing this out. We have rectified the typo (Line 325)

Reviewer 2 Report

Nnaji and colleagues discussed missed opportunities for vaccination (MOV) and associated factors among children in Cape Town, South Africa. The authors assessed 674 children and their caregivers and found that the MOV rate was 14.1%, most common in second dose of measles vaccine, and less common in first dose of rotavirus vaccine. Low level of education, decision making by parents are the main risk factor associated with the MOV.

The manuscript is important, I have some comments

a) Complication associated with MOV related to each vaccine should be considered and mentioned especially in South Africa to alert people about the risk of MOV.

b) Is there any previous data about MOV in South Africa? It is better to compare the MOV with the previous year in the same country.

c) The article should include recommendation for alternatives/ solutions for MOV in children.

Author Response

Comment: Nnaji and colleagues discussed missed opportunities for vaccination (MOV) and associated factors among children in Cape Town, South Africa. The authors assessed 674 children and their caregivers and found that the MOV rate was 14.1%, most common in second dose of measles vaccine, and less common in first dose of rotavirus vaccine. Low level of education, decision making by parents are the main risk factor associated with the MOV.

Response: We appreciate this encouraging and valuable feedback.

Comment: The manuscript is important, I have some comments

  1. a) Complication associated with MOV related to each vaccine should be considered and mentioned especially in South Africa to alert people about the risk of MOV.

Response: Thank you for this thoughtful comment. We had briefly discussed the implications of dose-specific prevalence of MOV in the previous version of the manuscript. We have now discussed the points in more detail (Lines 303 – 305 and 334 – 338).

Comment: Is there any previous data about MOV in South Africa? It is better to compare the MOV with the previous year in the same country.

Response: Thank you for this suggestion. We have added comparisons of our MOV prevalence finding with those from previous national and local studies. (Lines 314 – 320).

Comment: The article should include recommendation for alternatives/ solutions for MOV in children.

Response: We have discussed the various implications of our findings for policy and practice and have proposed various practical, evidence-based and potentially useful interventions/strategies for remedying MOV at facility and system-wide levels (Lines 311 – 451).

Reviewer 3 Report

This is an overall well-written manuscript. One minor revision should be noted. 
Given that COVID-19 vaccine is also widely used for children above the age of 5 years old, please also discuss the COVID-19 vaccine among children, especially the school-year children. 

Author Response

Comment: This is an overall well-written manuscript.

Response: Thank you for this positive remark.

Comment: One minor revision should be noted. Given that COVID-19 vaccine is also widely used for children above the age of 5 years old, please also discuss the COVID-19 vaccine among children, especially the school-year children.

Response: We appreciate this feedback. While we agree that COVID-19 vaccines have been considered for children under the age of 5, kindly note that our study focused on routine childhood immunisation and vaccines currently recommended for children in South Africa by the Extended Programme on Immunisation-South Africa (EPI-SA). We have however discussed the implications of the COVID-19 pandemic for assessing and remedying MOV (Lines 436 – 448).

Round 2

Reviewer 2 Report

No further comments.